# The Effect of Mono- and Di-Saccharides on the Microbiome of Dairy Cow Manure and Its Odor

**DOI:** 10.3390/microorganisms13010052

**Published:** 2024-12-31

**Authors:** John H. Loughrin, Getahun E. Agga

**Affiliations:** Food Animal Environmental Systems Research Unit, Agricultural Research Service, United States Department of Agriculture, 2413 Nashville Road, Suite B5, Bowling Green, KY 42101, USA; getahun.agga@usda.gov

**Keywords:** 16S rRNA, amplicon sequencing, animal manure, dimethyl disulfide, lactic acid bacteria, malodor, microbiome, *p*-cresol, skatole, sugar

## Abstract

In a previous experiment, we showed that the odor of *Bos taurus* manure slurries could be improved by anaerobic incubation with the sugars glucose, lactose, and sucrose. This improvement was due to reductions in the concentrations of malodorants, including dimethyl disulfide, *p*-cresol, *p*-ethylphenol, indole, and skatole, and a shift to the production of fruity esters, including ethyl butyrate and propyl propanoate. Due to large concentrations of lactic acid produced by the sugar-amended manure slurries, we inferred that lactic acid bacteria were involved in improving the manure slurry odor. Here, through 16S rRNA amplicon sequencing for microbiome analysis, we show that lactic acid bacterial growth was promoted by the addition of all three sugars. *Lactobacillus buchneri* and an unknown *Lactobacillus* sp. were the most prominent lactic acid bacteria stimulated by sugar addition. Lactobacillales were found only in trace abundances in unamended manure slurries. The relative abundance of orders such as Clostridiales, Bifidobacteriales, and Erysipelotrichales were not noticeably affected by sugar amendment. However, the disaccharides lactose and sucrose seemed to increase the relative abundance of *Bifidobacterium,* whereas the monosaccharide glucose did not. We conclude that lactic acid bacteria are the primary bacteria involved in improving odor in dairy cow manure slurries and present strategies to enhance their abundance in animal wastes.

## 1. Introduction

Recently, we investigated if increasing the carbon-nitrogen ratio of cow (*Bos taurus*) manure slurries by the addition of simple saccharides could improve their odor [1]. The results of the previous study indicated that the addition of simple sugars to dairy manure reduced malodorants by shifting the odor profile towards more favorable aliphatic esters. Furthermore, sugar amendment resulted in an acidic pH as a result of increased lactic acid production, which could be due to the enrichment of lactic acid bacteria by the added sugars. This lab-scale experiment was performed because it has been previously suggested that adding Jerusalem artichoke (*Helianthus tuberosus*) tubers rich in inulin polysaccharides to pig (*Sus domesticus*) diets reduced offensive odors, most likely due to a reduction in skatole concentration [2].

In two independent experiments, manure slurries were incubated with increasing concentrations of the sugars glucose, lactose, and sucrose [1]. It was found that regardless of the sugar used, increasing the sugar concentration reduced the concentration of aromatic malodorous compounds derived from the amino acids tryptophan and phenylalanine, such as *p*-cresol, *p*-ethylphenol, and skatole, as well as reducing the concentration of dimethyl disulfide (DMDS), another characteristic manure malodorant. The addition of sugars to the cow manure slurries also led to the production of fruit-like aliphatic esters such as ethyl butyrate and propyl propanoate.

Increasing the concentration of sugars significantly (analysis of variance [ANOVA] *p* < 0.05) decreased the concentration of aromatic malodorants in a dose-dependent manner, although low concentrations of sugars enhanced the production of DMDS (Figure 1). Higher concentrations of sugars significantly (ANOVA *p* < 0.05) enhanced the production of the aliphatic esters. The esters were not detected in control slurries.

As inferred by Farnworth et al. for the pig gut microbiome [2], we also surmised that the changes in the microbiota of the slurries could be responsible for the reduction of the malodorants. Lactic acid bacteria (LAB) are known to produce aliphatic esters, as in the fermentation of baijiu and dairy products [3,4,5]. Furthermore, fermentation of feed by *Lactobacillus* has been shown to reduce the malodor of pig feces [6]. Given that the addition of the sugars greatly enhanced the production of lactic acid, we hypothesized that the changes in the odor of manure slurries incubated with sugar could be by promoting the growth of lactic acid bacteria (LAB).

In any case, without sequencing data, this deduction was merely speculative. Here, we present 16S rRNA gene amplicon data that documents how incubation of cow manure slurries with increasing concentrations of sugars affects the microbiome of the slurries and discuss how the addition of carbohydrates to manure slurries could be used as a means of reducing malodorous emissions from animal production facilities.

## 2. Materials and Methods

### 2.1. Manure Incubations

Anaerobic incubations of the cow (*Bos taurus*) manure were described previously [1]. Briefly stated, 50 mL of manure was diluted to 100 mL with deionized water and placed in 150 mL crimp-top bottles equipped with rubber septa. Prior to the addition of the manure slurry, sugar was added to the vials as a dry powder. We relied on bacterial growth to obtain anaerobic conditions in the tightly sealed bottles. Two independent laboratory scale experiments were performed in a 3 × 5 factorial design, consisting of three sugars with five concentrations. The glucose and lactose concentrations used were 0, 6.5, 13.2, 26.3, or 65.8 g L^−1^ of sugar, corresponding to concentrations ranging approximately from 36 mM to 365 mM for glucose and 19 to 192 mM for lactose, thus keeping them at roughly equal concentrations when considered as units of monosaccharide. In the case of sucrose, the manure slurries were amended with 0, 12.5, 25, 50, or 125 g L^−1^ of sugar, corresponding to a range of 36.5 to 365 mM. The 0 g/L concentrations were used as control groups for each of the three sugar types. The sucrose experiments were conducted first using a higher concentration of sugar, and the lactose and glucose experiments were conducted later using lower concentrations. The individual bottles for each sugar concentration were incubated at 30 °C for 28 days. The incubation temperature was chosen to mimic ambient temperature under field conditions for a low-cost anaerobic digestion system. Each sugar concentration was conducted in duplicate in the two trials, resulting in four biological replicates. The same manure stock, stored at 4 °C, was used in the two trials.

### 2.2. DNA Extraction, Library Preparation and Sequencing

For microbiome analyses, composite samples were prepared by mixing equal aliquots from the four replicates for each of the five concentrations of the sugars, including controls, at the end of the 28-day incubation period. Samples were stored at −20 °C until processed.

The samples were processed and analyzed using the Microbiome Sequencing Service: 16S/ITS Amplicon Sequencing (Zymo Research, Irvine, CA, USA). DNA was extracted using ZymoBIOMICS^®^-96 MagBead DNA Kit or ZymoBIOMICS^®^ DNA Miniprep Kit, resulting in a 50 µL elution volume. The DNA samples were prepared for targeted sequencing with the Quick-16S™ Plus NGS Library Prep Kit using Quick-16S™ Primer Set V3-V4. The primers were custom-designed by Zymo Research (Irvine, CA, USA) to provide the best coverage of the 16S rRNA gene while maintaining high sensitivity. The sequencing library was prepared using an innovative library preparation process in which PCR reactions were performed in real-time PCR machines to control cycles and, therefore, limit PCR chimera formation. The final PCR products were quantified with qPCR fluorescence readings and pooled together based on equal molarity. The final pooled library was cleaned up with the Select-a-Size DNA Clean & Concentrator™ (Zymo Research), then quantified with TapeStation^®^ (Agilent Technologies, Santa Clara, CA, USA) and Invitrogen Qubit 1X DSDNA High-Sensitivity Assay Kits ^®^ (Thermo Fisher Scientific, Waltham, WA, USA). The ZymoBIOMICS^®^ Microbial Community Standard was used as a positive control for each DNA extraction and targeted library preparation. Negative controls (i.e., blank extraction control and blank library preparation control) were included to assess the level of bioburden carried by the wet lab process. The final library was sequenced on Illumina^®^ NextSeq 2000™ with a p1 (cat 20075294) reagent kit (600 cycles) (Illumina Inc., San Diego, CA, USA). The sequencing was performed with a 30% PhiX spike-in.

#### 2.2.1. Bioinformatics Analysis

Unique amplicon sequences, also known as amplicon sequence variants (ASVs), were performed inferred from raw reads using the Division Amplicon Denoising Algorithm (Dada2) pipeline [7]. Chimeric sequences were also removed with the Dada2 pipeline. Taxonomy assignment was performed using Uclust from Qiime v.1.9.1. Taxonomy was assigned with the Zymo Research Database, a 16S database that is internally designed and curated, as a reference.

Composition visualization, alpha diversity, and beta diversity analyses were performed with Qiime v.1.9.1 [8]. Other analyses such as heatmaps, taxa to sequence variant (Taxa2SV) decomposer, and principal coordinate analysis (PCoA) plots were performed with internal scripts.

#### 2.2.2. Quality Control Standards and Absolute Abundance Quantification of the 16S rRNA Gene

The resulting microbial composition of the ZymoBIOMICS^®^ Microbial Community Standard measured in this project and the theoretical composition of the standard gave very similar results (Figure 2A,B). A quantitative real-time PCR was set up with a standard curve to determine the 16S rRNA gene abundance in the samples (Figure 2C). The standard curve was made with plasmid DNA containing one copy of the 16S rRNA gene region prepared in 10-fold serial dilutions. The primers used were the same as those used in Targeted Library Preparation. The equation generated by the plasmid DNA standard curve was used to calculate the number of gene copies in the reaction for each sample. The PCR input volume (2 μL) was used to calculate the number of gene copies per microliter in each DNA sample.

The number of genome copies per microliter of DNA sample was calculated by dividing the gene copy number by an assumed number of gene copies per genome. The value used for 16S rRNA gene copies per genome is 4. The amount of DNA per microliter DNA sample was calculated using an assumed genome size of 4.64 × 10^6^ bp, the genome size of *Escherichia coli*. This calculation is shown below:Calculated Total DNA = Calculated Total Genome Copies × Assumed Genome Size (4.64 × 10^6^ bp) × Average Molecular Weight of a DNA bp (660 g/mole/bp) ÷ Avogadro’s Number (6.022 × 10^23^/mole).

## 3. Results and Discussion

### 3.1. Sequence Read Summaries and 16S rRNA Gene Abundance

The sugar experiment generated a total of 11 million raw reads with a mean of 735,112 reads ranging from 558,758 to 970,712 reads. Although the average total read counts in the controls (i.e., no added sugar) were higher than the samples amended with increasing sugar concentrations, there was no clear trend in the total raw reads by sugar concentration. While the highest total reads were observed in the control lactose and sucrose, its highest was observed at the 26.3 g L^−1^ glucose concentration (Table 1).

The qPCR for 16s rRNA gene abundance determination showed a linear relationship with the cycle threshold (Ct) values, and all samples had less than 15 Ct values (Figure 2D). The 16S rRNA gene abundance was significantly (*p* = 0.013) higher in the manure stock, and its abundance did not differ significantly between the sugar types (Figure 3A). This difference could be attributed to the fact that the manure stock was undiluted as opposed to the slurries in the sugar experiments, which were diluted by 50%. In addition, the sugar experiments were measured after 28 days of incubation in anaerobic conditions that can differentially reduce only aerobic bacteria in addition to nutrient depletion during this time. The 16S rRNA concentrations tended to decrease (*p* = 0.054) as the sugar concentrations increased (Figure 3B). This suggests that increasing concentrations of sugar reduces the total bacteria by enriching specific bacterial taxa over others, as also described below.

### 3.2. Microbiome of Manure Stock

The most abundant bacterial phyla of the manure stock were Bacillota (syn. Firmicutes) at 66.7%, Actinobacteria at 21.1%, and Pseudomonadota (syn. Proteobacteria) at 4.5% of amplicon sequence variant (ASV) counts, comprising over 95 percent of non-Archaeal ASVs (Appendix A). The most abundant orders of the manure stock were Clostridiales, Lactobacillales, and Corynebacteriales, followed by Erysipelotrichales and Bifidobacteriales (Figure 4A). Thus, over 83 percent of bacterial ASVs consisted of these five orders. Other prominent orders included Bifidobacteriales and Micrococcales in the phylum Actinomyceota, Pseudomonadales (phylum Pseudomonadota), Synergistota, and Bacteroidales (phylum Bacteroidota). This result was similar to that of other studies investigating the microbiome of cow manure [9,10,11]. Archaea occurred in 3.3% of ASV counts, of which 91.7% belonged to the order Methanomicrobiales and 8.3% belonged to the order Methanobacteriales.

For reasons that will be apparent later in the discussion, it is important to note that of the Lactobacilliales, the genus *Lactobacillus* comprised only 0.11% of ASVs in the manure slurry. Other species belonging to the genera, such as *Aerosphaera* at 2%, *Lactococcus* at 3%, and *Enterococcus* at 1.6%, and species belonging to the family Carnobacteriaceae at 4.1% of ASVs, were the most abundant Lactobacilliales.

### 3.3. Effect of Sugars on Manure Microbiome

The final pH of the control slurries averaged 5.80, 6.14, and 5.69 for glucose, lactose, and sucrose controls, respectively. At the highest concentrations of sugar supplementation, the slurries had final pHs of 3.38, 3.75, and 3.37 for glucose, lactose, and sucrose, respectively [1]. At higher concentrations of sugar addition, the most prominent short-chain fatty acid (SCFA) was lactic acid, regardless of which sugar was used. In the final control slurries, lactate occurred at an average of 44.5 and 960 mg L^−1^ for glucose and lactose incubation experiments and was not detected in the sucrose experiments. At the highest level of sugar supplementation, lactate occurred at average concentrations of 35,700, 29,300, and 44,900 mg L^−1^ for glucose, lactose, and sucrose experiments, which represented 83.4, 59.6, and 82.8 percent of identified SCFA [1].

The Shannon index estimates the entropy of a system by measuring the number and proportion of different species in a community [12]. The microbial diversity of the slurries, as estimated by this index, is presented in Figure 5A. While Shannon indices of microbial diversity in the control slurries declined relative to that of the manure stock, the indices showed steep declines in proportion to the amount of sugar added to the slurries. The Shannon indices were 3.33, 4.31, and 3.39 for glucose, lactose, and sucrose controls, respectively, whereas they were 2.16, 2.34, and 2.16 for the highest concentration of sugar addition. Species richness averaged 534, 732, and 631 for the control glucose, lactose, and sucrose incubates, respectively. However, it declined to 396, 456, and 380 for the lowest concentration of sugar addition and to 111, 195, and 145 for the highest concentration of sugar employed. Thus, the total number and diversity of species were negatively affected by sugar addition, likely due to the acidification of the manure slurries. Microbial diversity, as measured by other alpha diversity indices at the sample level, is presented in the Appendix A.

Beta diversity is a measurement of microbial diversity differences between samples. Figure 5B–E shows the 3-dimensional principal coordinate analysis (PCoA) plot created using the matrix of the paired-wise distance between samples calculated by the Bray–Curtis dissimilarity using unique amplicon sequence variants (ASVs), Bray–Curtis plots, weighted Unifrac plots, and unweighted Unifrac plots. The plots clearly show significant species diversity differences between the samples. The biggest cluster consisted of four samples that were from manure slurries amended with higher concentrations of lactose and sucrose (26.3 and 65.8 g L^−1^ for lactose and 50 and 125 g L^−1^ for sucrose). The next cluster with three samples was comprised of manure slurries incubated with no lactose or sucrose and amended with 6.5 g L^−1^ of lactose. The remaining samples were either not closely clustered or formed a cluster of two samples.

Microbial composition bar plots at all phylogenetic ranks and their associated relative abundance data are provided in the Appendix A. Heatmaps showing the microbial composition at each taxonomic level are shown in the Appendix A.

Manure slurries, whether incubated with sugar or not, were dominated by bacteria in the order Clostridiales and comprised 60% of all ASVs when averaged for all three sugars (Figure 4B). In unamended slurries, Clostridiales comprised 64% of ASVs in control slurries, followed by Erysiplotrichales at 11.3%, Bacialles at 6.2%, Bacteroidales at 6.1%, and the domain Archaea, consisting almost entirely of methanogens belonging to the orders Methanobacteriales (e.g., *Methanobrevibacter* spp.), Methanomicrobiales (e.g., *Methanocorpusculum* spp.), and Methanosarcinales (e.g., *Methanosarcina* spp.). The Archaea collectively comprised an average of 3.1% of ASVs in the control slurries and 5.3% in the lowest concentration of sugar concentration but were not detected in slurries containing higher concentrations of sugar. Since methanogenesis is generally reported to be inhibited at an acidic pH [13,14], the decrease in Archean relative abundance was not surprising.

Similarly to the Clostridiales, Erysiplotrichales abundance was not markedly affected by sugar addition. As Clostridiales and Erysiplotrichales are members of the phylum Bacillota, both are characteristic fermentative members of the gut microbiome [15] and are, therefore, likely to be tolerant of high concentrations of SCFAs. The relative abundance of another order belonging to the Bacillota (syn. Firmicutes), Bacillales, was somewhat decreased by the addition of sugar. Bacteroidales (phylum Bacteroidota) are also common fermenters in the gut microbiome and produce SCFAs, such as propionate, acetate, and succinate [16]. Nevertheless, the relative abundances of Bacteroidales decreased in proportion to the concentration of all three sugars.

Lactobacillales were present at relative abundances of 0.01, 2.70, and 0.07 percent of ASVs in the control slurries for glucose, lactose, and sucrose, respectively, and at relative abundances of 9.4, 23.8, and 7.7 percent of ASVs at the highest concentrations of sugar added. Much of this increase was due to an increase in the relative abundance of *Lactobacillus buchneri,* which was not detected in glucose or sucrose control slurries and comprised only 7.6 × 10^−5^ percent of ASVs in lactose control slurries. At the highest concentration of sugar addition, *L. buchneri* comprised 1.2, 17.7, and 0.18 percent of all ASVs for glucose, lactose, and sucrose slurries, respectively, and an unidentified *Lactobacillus* sp. accounted for 12.4, 74.9, and 2.3 percent of all *Lactobacillus* spp. for glucose, lactose, and sucrose, respectively. The unidentified *Lactobacillus* sp. was not detected in control slurries but accounted for zero percent of ASVs for control and 6.5 g L^−1^ glucose incubations, and 0.10, 2.50, and 5.67 percent of ASVs for 13.2, 26.3, and 65.8 g L^−1^ glucose incubations, 0, 0.02, 0.43, 1.83, and 3.26 percent of ASVs for 0, 6.5, 13.2, 26.3, and 65.8 g L^−1^ lactose incubations, and 0 percent of ASVs for control and 12.5 g L^−1^ sucrose incubations and 11.3, 7.0, and 6.9 percent for 25, 50, and 125 g L^−1^ sucrose incubations. Although in this paper we refer to *L. buchneri* as belonging to the genus *Lactobacillus*, it has been proposed to transfer it into the genus *Lentilactobacillus* [17]. *L. buchneri* is commonly used as an inoculant to improve the stability of silages [18].

Other Lactobacillales, which were prominent components of the manure microbiome, were either absent in the final control slurries or were present in very low percentages. *Enterococcus,* for instance, comprised 0.01, 0.46, 0.63, 0.02, and 0 percent of ASVs for 0, 6.5, 13.2, 26.2, and 65.8 g 8 g L^−1^ glucose incubations, 0.01, 1.11, 0.47, 0, and 0 percent for 0, 6.5, 13.2, 26.2, and 65.8 g 8 g L^−1^ lactose incubations, and 0 percent for all sucrose incubations except for 12.5 g L^−1^ of sucrose where they accounted for 0.05% of ASVs. Another family in the Lactobacillales, Carnobacteriaceae, was present in control lactose incubations at 1.68 percent of ASVs but was not detected in any of the slurries amended with lactose or any of the other manure slurries.

In the taxonomy to amplicon sequence variations (Taxa2ASV) analysis, a taxon of interest is decomposed into its unique sequences. Figure 6 shows the distribution of the top twelve most abundant unique sequences as bar plots for the genera *Lactobacillus* and *Clostridium*.

From this figure, it is clear that the growth of Lactobacilli, and in particular an unclassified *Lactobacillus* sp. and *L. buchneri*, were stimulated by the addition of all three sugars. *L. buchneri* was enriched at the highest concentration (68.8 g L^−1^) of lactose, and the unidentified *Lactobacillus* sp. was enriched when manure was incubated with sucrose at concentrations of 25, 50, and 125 g L^−1^. This suggests that the improvement in the manure odor was at least in part due to the growth of these bacteria.

As stated previously, Clostridiales were the most prominent order present in the manure slurries, whether amended with sugar or not (Figure 4). Over 20 *Clostridium* spp. were present in the manure slurries, and there was no apparent effect of sugar concentration on their abundance (Figure 4 and Figure 5). *C. celatum* was the most abundant species, and its relative abundance averaged 11.6, 14.1, 12.7, 18.2, and 17.5 percent of ASVs in slurries containing 0, 6.5, 13.2, 26.3, and 65.8 g L^−1^ of glucose, respectively, 5.2, 5.6, 5.3, 6.6, and 7.6 percent of ASVs in slurries containing 0, 6.5, 13.2, 26.3, and 65.8 g L^−1^ of lactose, respectively, and 13.2, 11.9, 10.5, 16.5, 17.0 percent of ASVs in slurries containing 0, 12.5, 25, 50, and 125 g L^−1^ of sucrose, respectively. In other words, *C. celatum* was differentially enriched when animal manure was incubated with glucose and sucrose, compared to lactose. This is interesting in that preserving fodders by ensiling them tends to reduce the growth of *Clostridium* spp. [19].

Several *Clostridium* spp. including *C. scatalogenes*, *C. nasuseum*, and *C. drakei,* are known to produce skatole and *p*-cresol [20,21]. *Lactobacillus* sp. strain 11201 has also been implicated in skatole production [22]. Another genus in the order Coriobacteriales, phylum Actinomyceota, *Olsenella*, has also been implicated in skatole biosynthesis [23]. *Olsenella* spp., however, occurring at an average of 0.12% in control slurries, were much less abundant than *Clostridium* spp., which accounted for an average of 12.4% of ASVs in control slurries. The abundance of *Olsenella* species also declined due to sugar addition, unlike was the case with *Clostridium*.

We also found that the relative abundance of *Bifiodobacterium pseudolongum* was enhanced by lactose and sucrose supplementation but not as much by glucose supplementation. Its relative abundance increased from 1.9% in control to 2.4% for 13.2 g L^−1^ of glucose but declined at higher concentrations and from 5.1% for lactose controls up to 29.5% for 65.8 g L^−1^ of lactose and from 0.1% of sucrose controls to 21.6% of ASVs in 25 g L^−1^ sucrose incubations. This is interesting because *Bifidobacterium* spp. has been shown to utilize plant- and milk-derived oligosaccharides in the human gut [24,25], and both disaccharides promoted its growth more so than the monosaccharide glucose. This is in contrast to the response of species in the genus *Bacteroides*, where there was not a marked change in the diversity of species in this genus in response to sugar supplementation except *B. propionicafaciens,* which was enriched at the highest two concentrations of lactose and sucrose. On the other hand, *B. graminisolvens* abundance was significantly higher in the lactose and sucrose control manures i.e., lactose and sucrose addition seemed to diminish the abundance of this species (Figure 7).

The differential response of the genera *Bifidobacterium* and *Bacteroides* to the addition of sugars illustrates the divergent effects of sugar addition on the manure microbiome. In the former’s case, glucose shows a slight tendency to enhance the growth of *B. pseudolongum* and *B. indicum*, whereas the disaccharides lactose and sucrose greatly enhance the relative abundance of *B. pseudolongum* (Figure 7). In the case of *Bacteroides*, the addition of sugars does not seem to have a pronounced effect on the relative abundance of the genus, but supplementation with either lactose or sucrose does seem to enhance the relative abundance of *B. propionicifaciens*.

Given the wide phylogenetic distribution of genes responsible for the biosynthesis of aromatic malodorants, it is unclear if the reduction in the concentrations of malodorants, such as skatole and *p*-cresol, was due to the decline in the population of malodor-producing bacteria or to changes in the environment (e.g., pH) which rendered the biosynthesis of malodors unfavorable. The slurry microorganisms responsible for the production of aliphatic esters, such as ethyl butyrate and propyl propionate, seem likely. However, previously, we speculated that LAB was responsible for the production of these esters given the large increase in lactic acid in the slurries when amended with sugars and the fact that LAB strains are known producers of esters in food products such as cheeses and other dairy products [3,4,5] as well as wines [26] and other fermented beverages [5,27], as well as meats [28]. Reductions in the concentration of aromatic malodorants and DMDS could be due to a reduction in the populations of bacteria responsible for their production, either due to direct microbial competition or, as is more likely, a reduction in the concentration of malodor producing bacteria caused by a drop in the pH of the slurries over time [1].

Nykänen et al. also found that the addition of the disaccharides maltose, lactose, or sucrose to pig waste lowered the pH and reduced the emissions of the volatile sulfur compounds hydrogen sulfide, methyl mercaptan, and dimethyl sulfide [29]. They also noted subjective improvements in the manure odor. While they also investigated the microbiome of the sugar-amended manure, they supplemented the slurry with *Lactobacillus plantarum* and *L. amylophilus* at 3- to 4-day intervals over an incubation period of 30 days.

Can et al. also studied the effect of adding glucose and *L. plantarum* to swine manure slurry and found reductions in pH and ammonia emission, reductions in the SCFA butyric, valeric, and *iso*-valeric acids but an increase in the concentration of acetic and propanoic acid [30]. In contrast to the studies of Nykänen and Can, we found that the addition of sugars alone stimulated the growth of *Lactobacillus* spp., obviating the need for bacterial inoculants.

## 4. Conclusions

Due to the results of our previous study [1], we speculated that LABs were responsible for the improvement in manure malodor, especially regarding ester production since LABs are known to produce both lactic acid and aliphatic esters [3,4,5,6,26,27]. Results presented here show that the relative abundance of LAB was greatly enhanced, which adds more weight to this supposition.

A number of issues with adding sugar to animal waste slurries were not addressed in this study, such as the effect sugar-amended manure application might have on soils and crops. Large amounts of sugar were added to the manure slurries to effect favorable changes in their odor profile. It is possible, however, that the addition of lesser concentrations of sugar over long periods of time could also improve manure malodors by promoting the growth of *Lactobacillus* spp. without adversely affecting the population of other bacteria capable of hydrolyzing more complex carbohydrates. This could perhaps be performed by using agricultural wastes rich in sugars or by the addition of carbohydrate-rich wastes such as stover, thereby avoiding the rapid lowering of the pH of the manure. It is also possible that sugar-enriched slurries could be used as inoculants for anaerobic digesters, manure storage pits, or lagoons to alter the microbiome and result in beneficial changes in their odor.

## Figures and Tables

**Figure 1 microorganisms-13-00052-f001:**
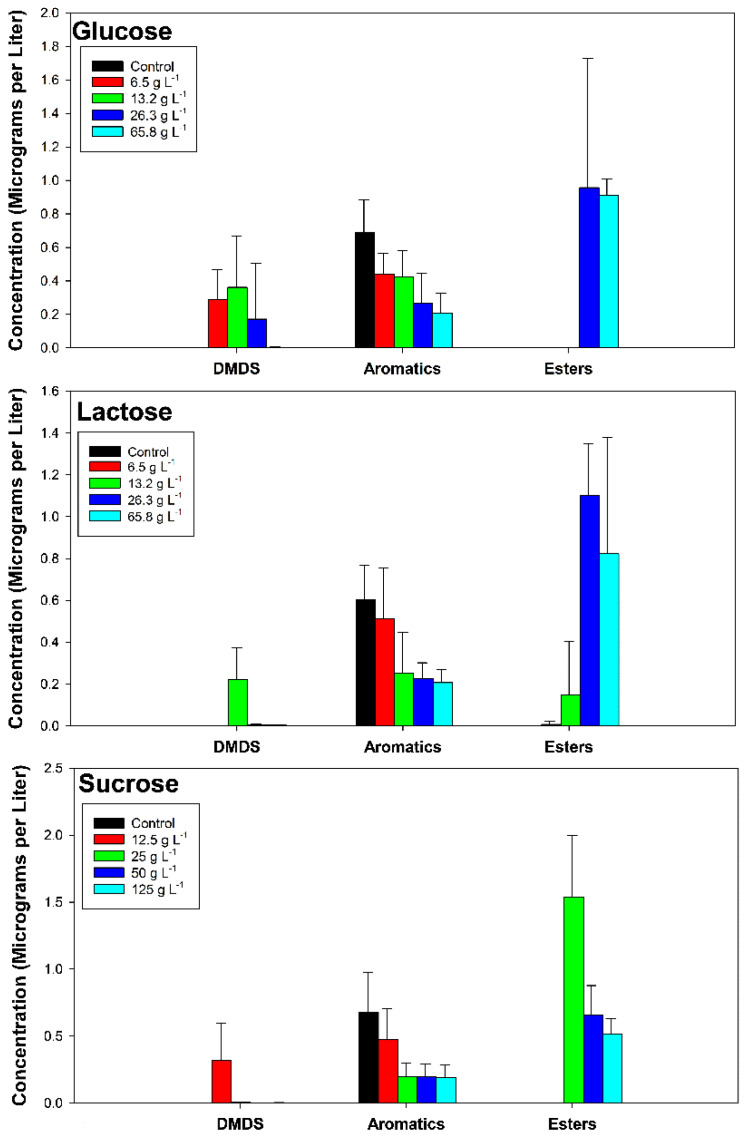
Odorants of cow manure slurries incubated with increasing concentrations of glucose, lactose, or sucrose assorted into biosynthetic categories (dimethyl disulfide [DMDS], aromatics, and esters). Data represent the mean of four determinations ± standard deviation of the mean. Adapted from reference [1].

**Figure 2 microorganisms-13-00052-f002:**
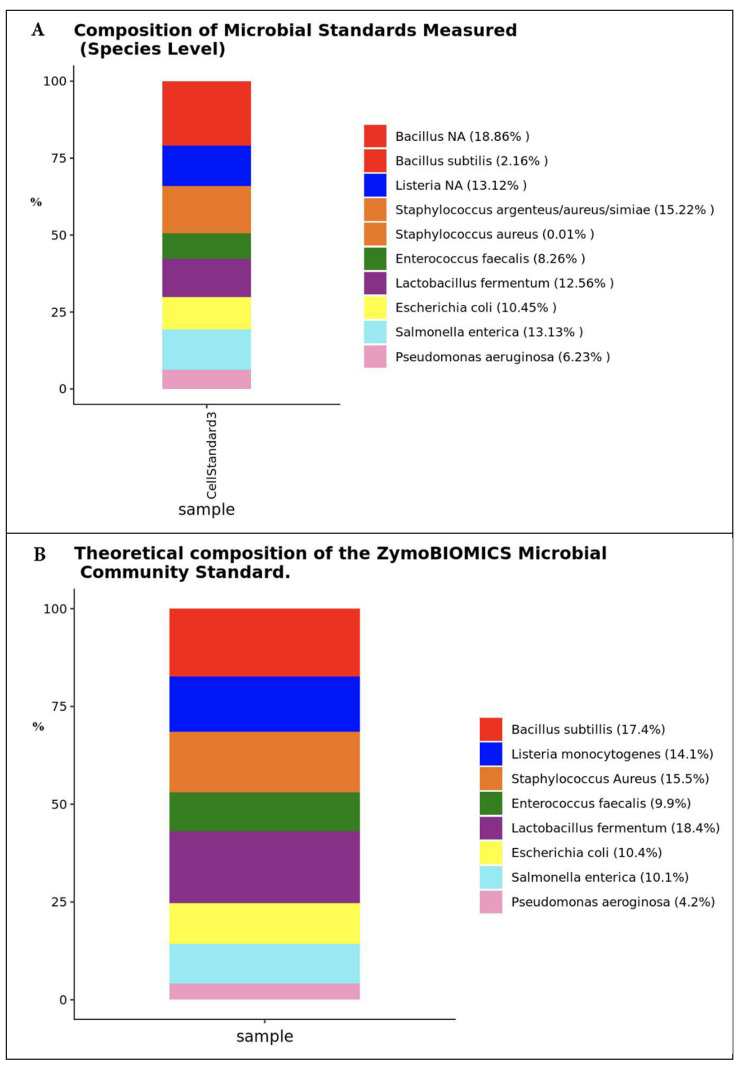
Quality controls and qPCR setup for microbial profiling and quantification of total bacteria. (**A**) Microbial composition of the ZymoBIOMICS^®^ Microbial Community Standard measured in this project; (**B**) the theoretical composition of the microbial community standard; (**C**) the absolute abundance standard curve used to generate total gene copies of 16S rRNA; (**D**) log_10_ 16S rRNA gene copies plotted against the cycle threshold (Ct) values.

**Figure 3 microorganisms-13-00052-f003:**
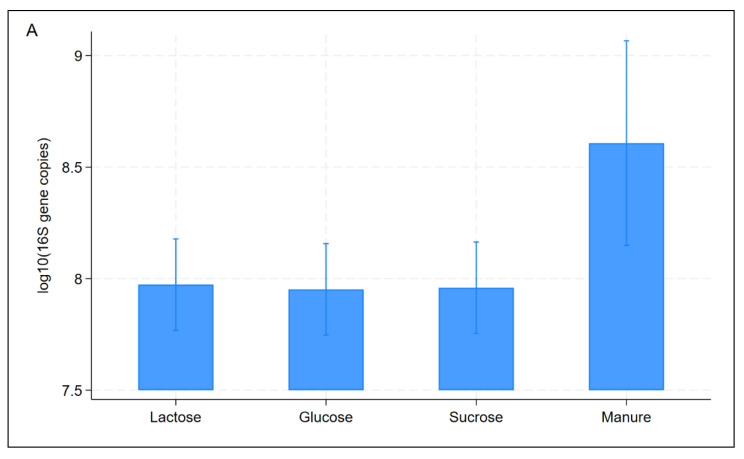
The abundance of 16S rRNA gene by sample type (**A**) and over the sugar concentrations (**B**).

**Figure 4 microorganisms-13-00052-f004:**
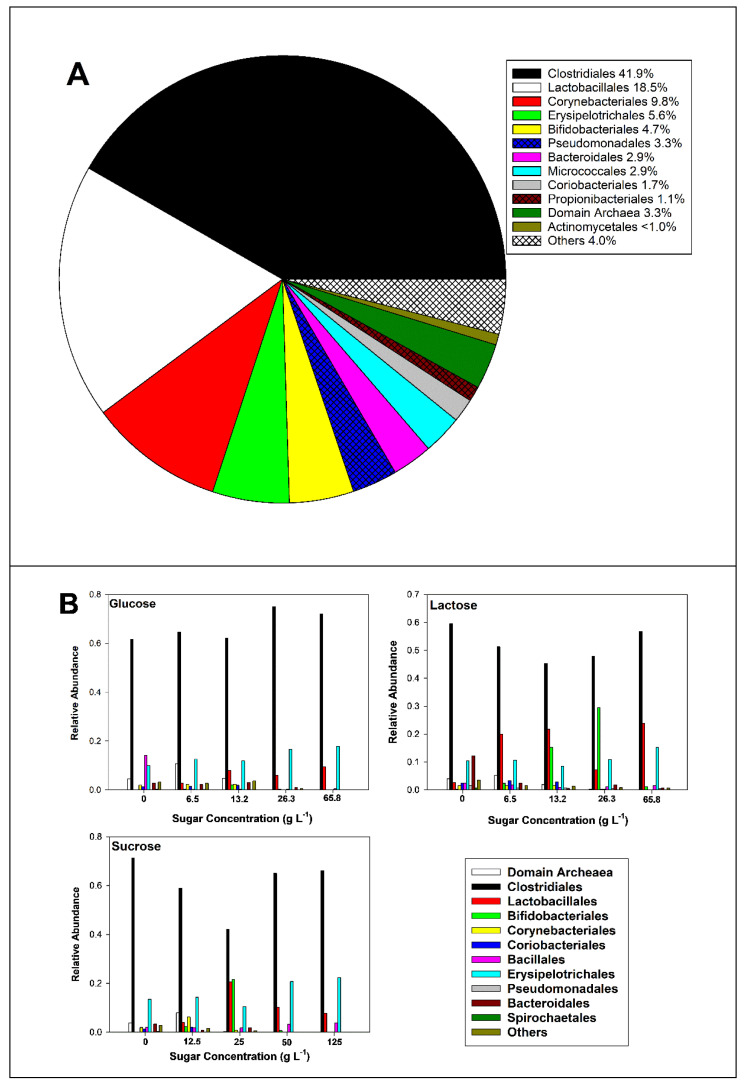
The most abundant bacterial orders and Archaea detected in the manure stock and slurries incubated with various concentrations of sugars. (**A**) Eleven most abundant bacterial orders and Archaea found in the manure stock used for sugar incubations; (**B**) changes in the relative abundance of Archaea and bacterial orders in response to sugar addition.

**Figure 5 microorganisms-13-00052-f005:**
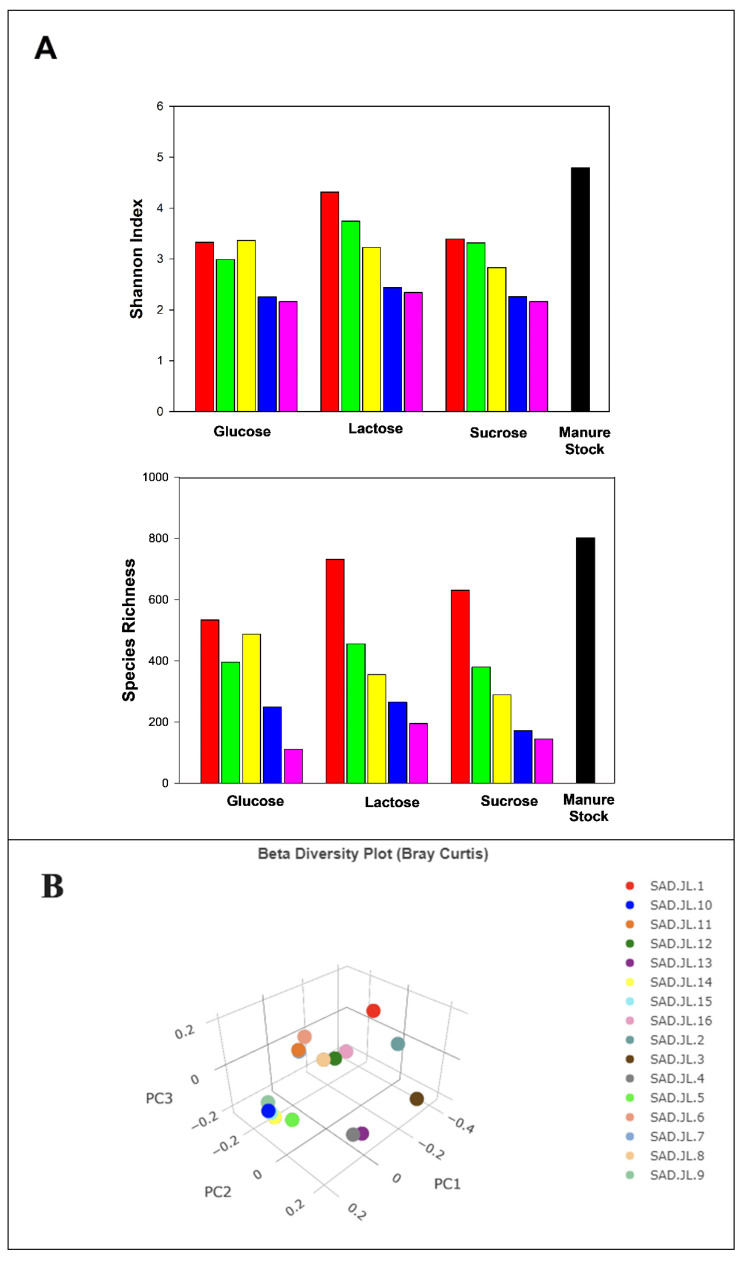
Microbial diversity of dairy manure slurries incubated with increasing concentrations of sugars. (**A**) Shannon indices and species richness of manure stock slurry and manure slurries incubated with increasing concentrations of sugar (0, 6.5, 13.2, 26.3, and 65.8 g L^−1^ for glucose and lactose; 0, 12.5, 25, 50, and 125 g L^−1^ for sucrose) aggregated by treatment group. Data represents 16S rRNA sequencing analysis of a composite sample of four replicates. Beta diversity plots (**B**–**E**): (**B**) 3-dimensional principal coordinate analyses (PCoA) using Bray–Curtis dissimilarity index; (**C**) Bray–Curtis dissimilarity index at the genus level; (**D**) weighted Unifrac based on ASV; and (**E**) unweighted Unifrac based on ASV. Labels: SAD JL 1-5: lactose control through lactose 65.8 g L^−1^; SAD JL 6-10: glucose control through glucose 65.8 g L^−1^; SAD JL 11-15: sucrose control through sucrose 125 g L^−1^; SAD JL 16: manure stock. PC1-PC3 represent the percentages of the total variations defined by the first three principal coordinates.

**Figure 6 microorganisms-13-00052-f006:**
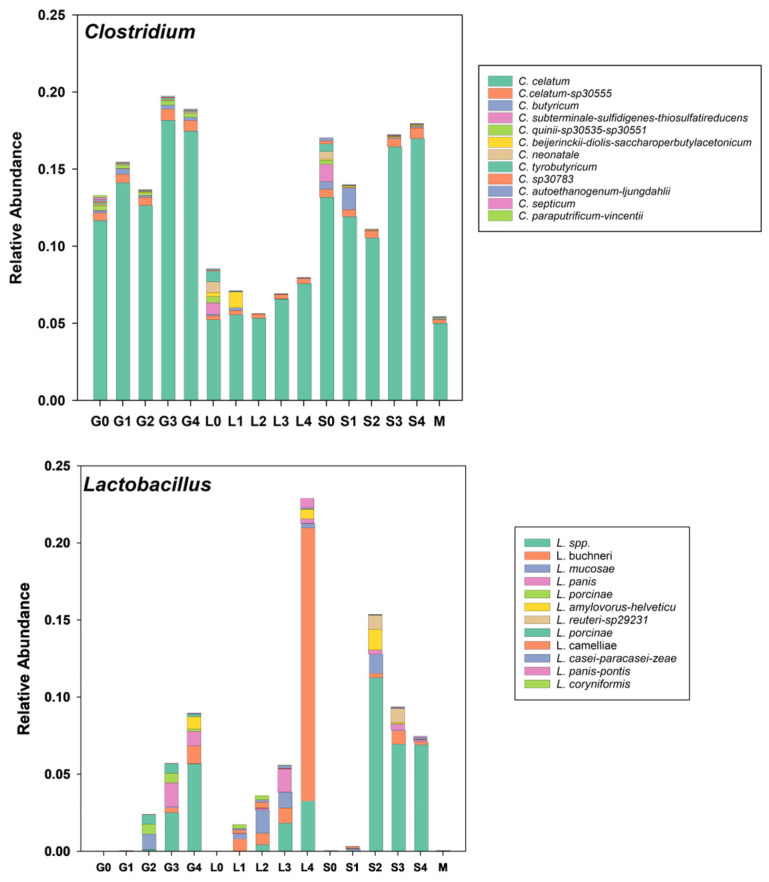
Twelve most abundant unique sequences for the genera *Clostridium* and *Lactobacillus* arranged in order of increased glucose, lactose, and sucrose concentrations plus in the manure stock used for sugar experiments.

**Figure 7 microorganisms-13-00052-f007:**
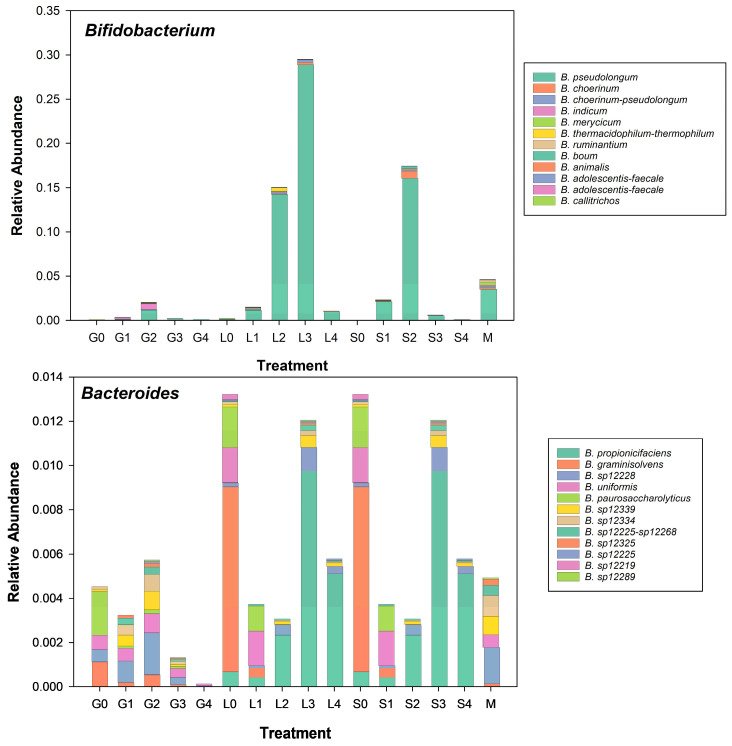
Twelve most abundant unique sequences for the genera *Bifidobacterium* and *Bacteroides,* arranged in order of increasing glucose, lactose, and sucrose concentrations plus in the manure stock used for sugar experiments.

**Table 1 microorganisms-13-00052-t001:** Read sequence processing summaries of the samples.

Sample ID	Raw Seqs (R1 + R2)	Trimmed Seqs (R1 + R2)	Dada2_Inferred	Chimera Seqs	Chimera Free Seqs	Unique Seqs	Seqs (After Size Filtration)	Final Unique Seqs
JL_1	830,432	747,490	358,580	37,439	321,141	2605	315,641	1251
JL_2	681,966	612,206	295,300	38,534	256,766	1661	253,220	760
JL_3	699,626	627,416	305,161	36,560	268,601	1305	266,082	627
JL_4	624,326	562,152	275,720	37,307	238,413	939	236,156	423
JL_5	694,890	625,824	307,446	54,099	253,347	806	251,343	316
JL_6	766,616	690,068	332,640	53,783	278,857	1973	274,440	889
JL_7	558,758	502,002	243,887	30,840	213,047	1315	210,032	651
JL_8	839,386	753,944	367,117	51,138	315,979	1633	312,578	821
JL_9	970,712	873,768	430,532	74,037	356,495	996	354,014	400
JL_10	657,314	591,086	291,481	55,114	236,367	465	235,430	202
JL_11	903,036	811,470	391,794	63,108	328,686	2463	322,958	1092
JL_12	657,834	590,022	285,863	44,025	241,838	1450	238,732	663
JL_13	664,628	597,048	292,499	36,607	255,892	1015	253,757	486
JL_14	776,662	699,276	344,356	57,250	287,106	705	285,581	295
JL_15	700,498	630,584	310,629	52,463	258,166	613	256,903	268
JL_16	592,992	531,834	248,965	25,349	223,616	3031	217,116	1353

## Data Availability

Raw sequence reads for all samples in this project have been deposited to the National Center for Biotechnology Information (NCBI) under a BioProject PRJNA1168161.

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
