# Peer review of "The Effect of Mono- and Di-Saccharides on the Microbiome of Dairy Cow Manure and Its Odor"

_microorganisms, 2024, doi:10.3390/microorganisms13010052_

Round 1

Reviewer 1 Report

Comments and Suggestions for Authors

Comments to Dairy cow manure slurry manuscript

The manuscript describes how the addition of disaccharides lactose and sucrose, but not the monosaccharide glucose, to cow manure slurries can increase the presence of two Lactobacilli species under anaerobic incubation. This procedure led to a shift to the production of fruity esters and a reduced production of malodorants. From these data, the authors include that lactic acid bacteria are the primary bacteria involved in reducing the malodorants.

There are two major questions concerning the procedure and the conclusion:

1.       How would the presence of sugars and their metabolized product lactic acid affect fertilization of the earth of the manipulated manure? The procedure leads to low pH of the manure (highly acidic) with consequent effect on agriculture earth and plant culture condition. This might also affect plant resistance to agricultural pests. These issues need to be discussed.

2.       If Lactobacillus is responsible for the reduced production of malodorants, would the external addition of these bacteria to the cow manure slurries be sufficient to reduce the malodorants?

Specific comments:

·         Line 16: First time a bacterial name is mentioned it should be presented in full name.  The bacterial name is also misspelled. It should be L. buchneri.

·         Please provide an experiment showing that lactic acid bacteria are indeed the reason for the reduced production of malodorants.

·         Line 29: The conclusion of Ref. 1 should be described here.

·         Line 34: It is unclear whether "these experiments" refer to the previous paragraph, or is related to the results of this paper. The introduction should solely describe what is known in literature. You should not only describe what you have previously observed, but combine this with data obtained from other research groups as well.

·         References should be added to paragraph 2 of the introduction.

·         Avoid using "we" in paragraph 2.

·         Figure 1 does not belong to introduction. If you have not shown this in the cited manuscript, you can add it to the result section. If it has been presented in reference 1, you should refer to this reference. Also, there are flaws in the figure. The number/L should be described as being in gram/L. (The unit is missing). The time of incubation is missing, and the incubation conditions are missing. There is a spelling mistake in the Y-axes: It should be "concentration" and not "concentariuon. Also, statistics are lacking.

·         An attempt for a hypothesis is written in lines 51-53, which should appear after comprehensive background information.

·         Lines 55-59: Is a kind of speculation, and should be part of the Discussion section.

·         The introduction has to be rewritten to provide comprehensive background information about manure and the known effects of sugars and other additives on the production of malodorants.

·         Why were lower concentrations of lactose used in comparison to sucrose and glucose? For lactose you considered the units of monosaccharides, but this was not considered for sucrose. Why?

·         How was the sugar added? As dry powder or in solution?

·         Line 77. There is a yellow outline that should not have appeared.

·         The sentence in lines 75-77 is not clear and should be better described. You did 4 experiments on two independent trials: It means that you tested the microbiota for each sample four times, and only two experiments from scratch were performed? Please clarify in the text.  

·         Line 80: four and not five concentrations were used for each sugar. Please correct the text.

·         Line 81: How long did you store the samples at four degrees? Can this storage affect microbial composition? Why were the samples not frozen until analysis?

·         Line 95: Spelling mistake. Correct to "Sensitivity".

·         Line 123: Correct to (Ct). Ct should be corrected throughout the text.

·         A statistic section is lacking in Method section

·         Line 128: 6 in 106 should be in superscript. (Please proofread your paper before submitting). The same for line 131.

·         Line 129: The bacterial name should be in italics.

·         Line 132: 32 in 1023 should be in superscript.

·         Line 137: Instead of 0 g/L you can state "no sugar".

·         Results and Discussion should be two separate sections to better emphasize what the new data presented in this paper is. The Discussion should discuss the main findings and integrate it with the literature.

·         The letters in the figures should be made readable.

·         Figure 3: The height and width of subfigures A and B (the A and B labels should be added and defined in the legend) should be similar in the two subfigures. And the letters in the figures should be readable. Please add a label to the Y-axes.

·         How can you explain why the 16S rRNA gene abundance was significantly higher in the manure stock?

·         If the 16S rRNA concentrations decrease as the sugar concentrations increase, it suggests that sugar-consuming bacteria have a negative effect on the other bacteria. Please comment on it.

·         Line 161: Refer specifically to the Supplementary Figure. Not sufficient to write "supplementary material".

·         Should the Bacterial phyla be written in italics?

·         Line 185: For a comparison of the three sugars effect on manure slurries, the same slurry should have been used for all sugars. So how can it be that the lactate was different in the different controls? What was the initial lactate concentration in the manure slurries?

·         What were the butyrate, propiniate concentrations?

·         Line 190: Please write "in" instead of "as".

·         Line 189: The meaning of Shannon index should be defined.

·         Figure 6 lacks statistics and number of replicates should be stated in the legend. Also, the different colors of the bars are not defined. Please define them.

·         The entire figure 7 should be as in subfigure A. The labeling should be better presented with true treatments. The color ranges should be closer for each sugar presented for better presentation. The manure stock should be in black color to easily distinguish it from the others. PC1, PC2 and PC3 parameters should be defined in the figure legend. The labels in the black subfigures are unreadable. The text should better emphasize what you can conclude from Figure 7. The numbers in A should be subsequent (in present form it is. 1, 10-16, 2-9).

·         Lines 220-221 have unnecessary yellow labeling.

·         Line 230: Mention here the name of the methanogenic bacteria that are reduced.

·         Figure 8 lacks statistics. Add asterisk to significant changes. The red bar should it be Lactobacillales? (written Lactobacilles).

·         Instead of using your code index of SAD.JL.1-16, the true treatments should be stated.

·         For the png figures attached to the manuscript, the identity of the different colors should be defined.

·         Figure 5 should also show the distribution of treated samples.

·         Line 327: B. pseudolongum should be in italics.

·         Figure 9: You need to state what the Y-axis number is presenting (ratio or percentage). The figure itself seems to be homemade as the columns are not straight. The same for figure 10. The labeling in these two figures of the different treatment groups is much better than the SAD.JL.1-16 labeling, and should therefore be applied to all the other figures. In the figure legends, all bacterial names should be in italics.

·         A major problem in the result section is the extensive description of percentages, which makes the sentences heavy, and difficult for the reader to draw conclusion. The text should better emphasize which changes are statistically significant, and the figures should present the percentage data. Also +/- variations should be added to the percentages, and what is statistically significant.   

·         The conclusion is general. A broader conclusion of the main findings of the data presented in the manuscript should be provided.

·         There is a total lack of discussion with integration with known data from the literature.

Author Response

The manuscript describes how the addition of disaccharides lactose and sucrose, but not the monosaccharide glucose, to cow manure slurries can increase the presence of two Lactobacilli species under anaerobic incubation. This procedure led to a shift to the production of fruity esters and a reduced production of malodorants. From these data, the authors include that lactic acid bacteria are the primary bacteria involved in reducing the malodorants.

We appreciate the reviewer for their nice summary of the work and their constructive comments and suggestions.

There are two major questions concerning the procedure and the conclusion:

  1. How would the presence of sugars and their metabolized product lactic acid affect fertilization of the earth of the manipulated manure? The procedure leads to low pH of the manure (highly acidic) with consequent effect on agriculture earth and plant culture condition. This might also affect plant resistance to agricultural pests. These issues need to be discussed.

We feel that the acidic nature of the sugar amended manure as a means to control malodor may make it impractical to use as  organic fertilizer. We briefly discussed in the Conclusion how this might be circumvented by other means such as adding agricultural waste rich in polysaccharides.  We feel that discussing the effect that the acidic manure would have on plants and soils would be purely speculative but is an interesting subject. However, the main objective of this study was to reduce the odor emitted from animal manure storage facilities, by testing the potential of adding simple sugars. The effect of sugar treatment on soil fertility and plant growth requires a separate study and it was beyond the scope of the present study.

  1. If Lactobacillus is responsible for the reduced production of malodorants, would the external addition of these bacteria to the cow manure slurries be sufficient to reduce the malodorants?

Yes, this was the goal of the microbiome analysis: to identify bacterial taxa that is differentially enriched by sugar addition. This would require another study by itself to figure out the added Lactobacillus bacteria could be maintained in the manure without adding sugars.

Specific comments:

  • Line 16: First time a bacterial name is mentioned it should be presented in full name.  The bacterial name is also misspelled. It should be L. buchneri. We corrected the spelling.
  • Please provide an experiment showing that lactic acid bacteria are indeed the reason for the reduced production of malodorants. Based on the previous article (https://doi.org/10.3390/environments11070145), since the concentration of lactic acid was increased by incubation with sugar, we speculated that lactic acid bacteria were responsible for the improvement in manure odor, particularly in regard to ester production. The purpose of this paper was to determine how the microbiome of the manure was affected by sugar addition, and especially how the relative abundance of lactic acid bacteria was affected. We did find that the relative abundance of lactic acid bacteria was greatly increased. We added a paragraph to the Conclusions stressing this finding.
  • Line 29: The conclusion of Ref. 1 should be described here.

We inserted the main conclusions of the previous study.

  • Line 34: It is unclear whether "these experiments" refer to the previous paragraph, or is related to the results of this paper. The introduction should solely describe what is known in literature. You should not only describe what you have previously observed, but combine this with data obtained from other research groups as well. We added the reference to Loughrin et al. on the sentence to make it clear we were referring to our published research. Findings from the literature were already described in the next paragraphs. The preceding paragraphs were meant to describe the experimental studies and previous finding with respect to physical and chemical analyses.
  • References should be added to paragraph 2 of the introduction. We added the reference to our previous research
  • Avoid using "we" in paragraph 2. We deleted the words we.
  • Figure 1 does not belong to introduction. If you have not shown this in the cited manuscript, you can add it to the result section. If it has been presented in reference 1, you should refer to this reference. Also, there are flaws in the figure. The number/L should be described as being in gram/L. (The unit is missing). The time of incubation is missing, and the incubation conditions are missing. There is a spelling mistake in the Y-axes: It should be "concentration" and not "concentariuon. Also, statistics are lacking.

Figure 1 presents a summary of our previous paper published in Environments. It is acceptable in the Introduction explaining the rationale for conducting the present study. I apologize for the errors in this figure, apparently, I need a pair of reading glasses. My third eye also missed it. Now the figure is corrected. A formal statistical analysis was presented in the previous publication. Statistical significances based on ANOVA are added in the text.

  • An attempt for a hypothesis is written in lines 51-53, which should appear after comprehensive background information. Lines 55-59: Is a kind of speculation, and should be part of the Discussion section. These sentences were moved as suggested.
  • The introduction has to be rewritten to provide comprehensive background information about manure and the known effects of sugars and other additives on the production of malodorants. There is not much background information on this topic and the literature is limited.
  • Why were lower concentrations of lactose used in comparison to sucrose and glucose? For lactose you considered the units of monosaccharides, but this was not considered for sucrose. Why? The sucrose experiments were conducted first using a higher concentration of sugar and the lactose and glucose experiments were conducted later using lower concentrations,
  • How was the sugar added? As dry powder or in solution? We added the line “Prior to addition of the manure slurry, sugar was added to the vials as a dry powder.” In the section 2.1.

  • Line 77. There is a yellow outline that should not have appeared. I’m sorry but I don’t see this outline on my word or pdf versions of the submitted manuscript.

  • The sentence in lines 75-77 is not clear and should be better described. You did 4 experiments on two independent trials: It means that you tested the microbiota for each sample four times, and only two experiments from scratch were performed? Please clarify in the text.  For budgetary reasons, the microbiomes were compared by mixing all four replicates into one composite sample as stated at the beginning of section 2.2.
  • Line 80: four and not five concentrations were used for each sugar. Please correct the text.

We changed this to read, “For microbiome analyses, composite samples were prepared by mixing equal aliquots from all four replicates for each of the five concentrations of the sugars including control at the end of the 28-day incubation period.

  • Line 81: How long did you store the samples at four degrees? Can this storage affect microbial composition? Why were the samples not frozen until analysis? Sorry for the oversight. We asked our technical team and corrected it. Samples were stored frozen at -20C.
  • Line 95: Spelling mistake. Correct to "Sensitivity". Changed
  • Line 123: Correct to (Ct). Ct should be corrected throughout the text. Changed.
  • A statistic section is lacking in Method section.

We did not perform any statistical comparisons of the microbiome since these were composite samples of all four replicates for all five concentrations of sugars including the control.

  • Line 128: 6 in 106 should be in superscript. (Please proofread your paper before submitting). The same for line 131. We corrected these plus 1023. (Line 152)
  • Line 129: The bacterial name should be in italics. We corrected this.
  • Line 132: 32 in 1023 should be in superscript. Done. (line 156)
  • Line 137: Instead of 0 g/L you can state "no sugar". We changed this to no added sugar. (Line 176).
  • Results and Discussion should be two separate sections to better emphasize what the new data presented in this paper is. The Discussion should discuss the main findings and integrate it with the literature.
  • The letters in the figures should be made readable. The Figures resolutions were improved. In addition to inserting the figures in the appropriate location in the text as a place-holder, we also attached them individually so the journal production department can rearrange them for a better quality. We combined some of the graphs as a panel into one figure. Example, we combined the original figures 2 and 3, into Figure 3 with panels A-D.
  • Figure 3: The height and width of subfigures A and B (the A and B labels should be added and defined in the legend) should be similar in the two subfigures. And the letters in the figures should be readable. Please add a label to the Y-axes.

These are done. The figures and their Y-axis are labelled, and the text are now more legible. The figures have the same scale on their y-axis in 25 increments. However, since the graphs were produced by the company, we were unable to modify the width (X-axis).

  • How can you explain why the 16S rRNA gene abundance was significantly higher in the manure stock? The manure stock was undiluted, the slurries were diluted by 50%. Nutrient depletion during incubation and the anaerobic condition could also affect abundances. These were included in the text.
  • If the 16S rRNA concentrations decrease as the sugar concentrations increase, it suggests that sugar-consuming bacteria have a negative effect on the other bacteria. Please comment on it. We added a brief statement here. We commented on this at length in discussing the Shannon index and species richness in Figure 6. We also added the sentence “Thus, the total number and diversity of species was negatively affected by sugar addition likely due to acidification of the manure slurries.” At the end of paragraph two, section 3.3.(Line 246-247)      

Line 161: Refer specifically to the Supplementary Figure. Not sufficient to write "supplementary material". We changed this to “Microbial diversity as measured by other alpha diversity indices at sample level is presented in the supplementary files (Supplementary Figure Alpha Diversity Bar Plots.docx).” We also changed it to Supplementary files.

  • Should the Bacterial phyla be written in italics? No, only at the genus and species level.
  • Line 185: For a comparison of the three sugars effect on manure slurries, the same slurry should have been used for all sugars. So how can it be that the lactate was different in the different controls? What was the initial lactate concentration in the manure slurries? These were final measurements, which is now inserted for clarity. The same manure stock was used for all trials. We did not measure SCFA in the initial manure stock.
  • What were the butyrate, propiniate concentrations? This can be seen in the Environments paper.
  • Line 190: Please write "in" instead of "as". We changed this.
  • Line 189: The meaning of Shannon index should be defined. We gave a brief overview of the intent of the Shannon Index.
  • Figure 6 lacks statistics and number of replicates should be stated in the legend. Also, the different colors of the bars are not defined. Please define them. This, and other figures represent an analysis of a composite sample of all four replicates. The sentence “Data represents 16S rRNA sequencing analysis of a composite sample of all four replicates.” Was added to the caption. Descriptions of the colors for the increasing sugar concentrations are added to the captions.
  • The entire figure 7 should be as in subfigure A. The labeling should be better presented with true treatments. The color ranges should be closer for each sugar presented for better presentation. The manure stock should be in black color to easily distinguish it from the others. PC1, PC2 and PC3 parameters should be defined in the figure legend. The labels in the black subfigures are unreadable. The text should better emphasize what you can conclude from Figure 7. The numbers in A should be subsequent (in present form it is. 1, 10-16, 2-9). Since the bioinformatics analyses were done by a commercial company, we do not have access to the pipeline to make changes. The generated figures could not be edited.
  • Lines 220-221 have unnecessary yellow labeling. I cannot see this in my pdf or word copy of the manuscript. If it persists, the journal production department can remove during proof reading.
  • Line 230: Mention here the name of the methanogenic bacteria that are reduced. In order to reduce the complexity of the text, we added the names of representative genera from each order of methanogenic Archaea. Information added.
  • Figure 8 lacks statistics. Add asterisk to significant changes. The red bar should it be Lactobacillales? (written Lactobacilles). Again, we state that no meaningful statistics can be performed on this data. They do show distinct changes in the relative abundance of bacterial orders and Archaea in response to sugar addition, however. We corrected this figure to read Lactobacillales and apologize for the error.
  • Instead of using your code index of SAD.JL.1-16, the true treatments should be stated. These were retained as originally labelled and sent out for sequencing and bioinformatics services.
  • For the png figures attached to the manuscript, the identity of the different colors should be defined. These are modified and replaced.
  • Figure 5 (changed to Figure 4) should also show the distribution of treated samples. This figure represents the microbiome of the manure stock and can be used for reference to the sugar experiments showing how sugars affected the microbiome of the manure. The two profiles are now shown in one graph (as 4A and B) for ease of visualization). Figure 5 (current) does do a comparison of the manure stock versus the treated samples as regards the Shannon diversity index and species richness.
  • Line 327: B. pseudolongumshould be in italics.  Done.

We corrected this.

  • Figure 9: You need to state what the Y-axis number is presenting (ratio or percentage). The figure itself seems to be homemade as the columns are not straight. The same for figure 10. The labeling in these two figures of the different treatment groups is much better than the SAD.JL.1-16 labeling, and should therefore be applied to all the other figures. In the figure legends, all bacterial names should be in italics. All bacterial names at the genus and species levels are in italics but has been corrected for figure 10. The graphs were prepared in Sigmaplot 15 and the awkward stacking of the stacked bars is a glitch in the program. We changed the border of the bars to hairline thickness and this reduced the awkward appearance of the bars somewhat. We changed the y-axis in Figures 8, 9 and 10 to read Relative Abundance. Note the figure numbers were changed in this revision as we complied  several similar figures into one.
  • A major problem in the result section is the extensive description of percentages, which makes the sentences heavy, and difficult for the reader to draw conclusion. The text should better emphasize which changes are statistically significant, and the figures should present the percentage data. Also +/- variations should be added to the percentages, and what is statistically significant.   Since this is data compared from a composite sample of all four replicates, we cannot do any meaningful statistics on the data. Still the shifts in the microbiome composition are clearly related to sugar addition. When discussing the microbiome it is impossible to discuss shifts in its composition without referring to percentages.
  • The conclusion is general. A broader conclusion of the main findings of the data presented in the manuscript should be provided. We expanded the conclusion . We added a paragraph to the Conclusions emphasizing that the proportion of lactic acid bacteria were enhanced by sugar addition.  
  • There is a total lack of discussion with integration with known data from the literature. There is not extensive literature on this subject but did include some more references at the end of the discussion.

Reviewer 2 Report

Comments and Suggestions for Authors

The authors studied the effects of adding various sugar concentrations on the microbial population and diversity in dairy cow manure. The study is crucial for identifying the best storage or disposal conditions for dairy cow manure that prevent further environmental pollution and odor emission.

Comments:

- Is the addition of pure sugar to control the odor of dairy cow manure feasible and applicable on an industrial scale? It might be better to study the co-digestion of dairy cow manure with other waste resources to provide the necessary nutrients, including sugars, to foster a microbial community that can eliminate odors during manure storage.

- What is the scientific rationale for choosing anaerobic incubation at 30°C for the tests? This temperature is suitable for psychrophilic anaerobic digestion, which is known for its very low methanogenic activity compared to mesophilic and thermophilic temperatures. The authors should provide scientific evidence to justify the use of 30°C.

- The experimental designs are very complex, and the authors have not described them clearly.

- Employing standard scientific terminology is essential to inform readers about how the tests were designed and conducted. Terms like 'manure stock,' 'unamended,' and 'amended trials' are confusing and should be replaced with standard terminology found in the literature.

-  It appeared that all the trials were under acidic conditions with very low pH values, which indicates that methanogenic activity was implausibly low. Therefore, it seems the differences in chemical compositions between the trials were due to varied chemical reactions rather than biological reactions, owing to the addition of different sugars and sugar concentrations.

- The presentation of the results and analysis is unclear to the reader and difficult to understand. The authors need to simplify how they present their results. A better graphical representation of the results is necessary to convey the outcomes of the experiments, rather than an overly lengthy and textual discussion of the results.

- A comparison of the results of this study with similar studies in the literature is not presented in the discussion of the results.

- Describing the aims and objectives of the study is crucial and should be detailed in a section of the introduction. Subsequently, outlining the outcomes of the study in the conclusion section of the manuscript is vital to demonstrate its success and achievements.

Comments on the Quality of English Language

na

Author Response

Is the addition of pure sugar to control the odor of dairy cow manure feasible and applicable on an industrial scale? It might be better to study the co-digestion of dairy cow manure with other waste resources to provide the necessary nutrients, including sugars, to foster a microbial community that can eliminate odors during manure storage.

We stated that the addition of sugars at this scale to manure was not practical and suggested that perhaps addition of agricultural wastes high in sugars or other complex carbohydrates might be a solution to this problem. The aim of this study was limited, however, simply how the sugars used in the referenced previous paper by Loughrin et al. affected the microbiome of the manure slurries.

- What is the scientific rationale for choosing anaerobic incubation at 30°C for the tests? This temperature is suitable for psychrophilic anaerobic digestion, which is known for its very low methanogenic activity compared to mesophilic and thermophilic temperatures. The authors should provide scientific evidence to justify the use of 30°C.

The justification was simply to have a defined temperature for the incubations. Thirty degrees was chosen simply because this was warm enough to complete the incubations in a timely manner and is not unreasonable temperature to mimic environmental conditions on a farm.   

- The experimental designs are very complex, and the authors have not described them clearly.

We described the experimental conditions in detail in the previously published Environments MDPI paper by Loughrin et al. (2024). Readers can easily access this material.  In this paper we emphasized the molecular analyses in detail.  We modified the text for clarity as well.

- Employing standard scientific terminology is essential to inform readers about how the tests were designed and conducted. Terms like 'manure stock,' 'unamended,' and 'amended trials' are confusing and should be replaced with standard terminology found in the literature.

We changed the terminology throughout the manuscript to make it more consistent

-  It appeared that all the trials were under acidic conditions with very low pH values, which indicates that methanogenic activity was implausibly low. Therefore, it seems the differences in chemical compositions between the trials were due to varied chemical reactions rather than biological reactions, owing to the addition of different sugars and sugar concentrations. We discussed the effect of sugar addition on methanogens in this paper and in reference #1. The effect was marked and there was a plausible affect on methane concentration as outlined in Loughrin et al. (2024). We are not sure what is meant by chemical reactions versus biological reactions since methane production, SCFA production and odor production are all the result of biochemical reactions.

- The presentation of the results and analysis is unclear to the reader and difficult to understand. The authors need to simplify how they present their results. A better graphical representation of the results is necessary to convey the outcomes of the experiments, rather than an overly lengthy and textual discussion of the results. We attempted to improve the quality of the graphics and make extensive use of them. The length of the discussion due to the nature of describing microbiome composition involves at length discussion of proportions and percentages but we tried to do this in relatively short passages with liberal graphical representation.  

- A comparison of the results of this study with similar studies in the literature is not presented in the discussion of the results. We expanded the Introduction to make the aims clearer and added some references to the Results and Discussion. The literature on the use of sugars to modify manure odor is not extensive.

- Describing the aims and objectives of the study is crucial and should be detailed in a section of the introduction. Subsequently, outlining the outcomes of the study in the conclusion section of the manuscript is vital to demonstrate its success and achievements. We reworded and expanded the Introduction of the paper to more adequately describe the aims and objectives of the study and added an iteration of this in the Conclusions.

Round 2

Reviewer 1 Report

Comments and Suggestions for Authors

The manuscript has been improved. There are only some few issues that need to be addressed:

Lines 47 and 49: If it is statistically significant the p  value should be lower and not greater than 0.05.

Section 2.1: Please add how you kept the incubation anaerobic.

Line 149: Something should be added after “and”?

Figure 5A: It would be preferable to add the sugar concentrations in the X-axis, such that the reader should not guess what is what. It would also be easier if the color of the bar columns where the same for each concentration.

Line 375: [28] appears between two “ands”. Please add the reference number immediately after “et al.” and delete the excess “and”. The same for [29] should be added after “Can et al.”.

Some words on the limitation of the study and its application in agriculture should be added.

Author Response

We thank you for your helpful comments.

The manuscript has been improved. There are only some few issues that need to be addressed:

Lines 47 and 49: If it is statistically significant the p  value should be lower and not greater than 0.05. Done

Section 2.1: Please add how you kept the incubation anaerobic. We added this on lines 74-75. 

Line 149: Something should be added after “and”? Deleted

Figure 5A: It would be preferable to add the sugar concentrations in the X-axis, such that the reader should not guess what is what. It would also be easier if the color of the bar columns where the same for each concentration. We changed the colors of the bars to be the same for each concentration. The graph became too busy adding the concentrations to the x-axis but explained the order of increasing concentrations in the figure legend.

Line 375: [28] appears between two “ands”. Please add the reference number immediately after “et al.” and delete the excess “and”. The same for [29] should be added after “Can et al.”. We fixed this.

Some words on the limitation of the study and its application in agriculture should be added. We added a few words on these limitations.

Reviewer 2 Report

Comments and Suggestions for Authors

NA

Author Response

We thank you for your help with this manuscript.

Round 3

Reviewer 1 Report

Comments and Suggestions for Authors

The manuscript has been improved and is now acceptable for publication.